# Cardiovascular Risk Factors in Hospital Workers during the COVID-19 Pandemic: A Hospital-Based Repeated Measures Study

**DOI:** 10.3390/ijerph192316114

**Published:** 2022-12-01

**Authors:** Mao-Hung Liao, Ying-Ching Lai, Chih-Ming Lin

**Affiliations:** 1Superintendent Office, Yonghe Cardinal Tien Hospital, New Taipei 234, Taiwan; 2Department of Medical Affair, Yonghe Cardinal Tien Hospital, New Taipei 234, Taiwan; 3Department of Healthcare Information and Management, Ming Chuan University, Taoyuan 333, Taiwan

**Keywords:** pandemic, health care workers, cardiovascular diseases, metabolic syndrome, work stress

## Abstract

Although many studies have investigated burnout, stress, and mental health issues among health care workers (HCWs) during the COVID-19 pandemic, few have linked these relationships to chronic physiological illnesses such as cardiovascular diseases. This study assessed changes in cardiovascular risk factors in HCWs and other hospital workers during the COVID-19 pandemic and identified vulnerable groups at a higher risk of increased adverse cardiovascular conditions. Five hundred and fourteen hospital employees ≥ 20 years of age underwent physical examinations and laboratory testing once before and once after the first wave of the pandemic in Taiwan during 2020 and 2021. Their sociodemographic characteristics and cardiovascular risk factors, including blood pressure, blood biochemical parameters, and body mass index, were collected. The differences between pre- and post-pandemic measurements of their biophysical and blood biochemical parameters were analyzed using pairwise tests. The post-pandemic increases in their parameter levels and cardiovascular risk as a function of underlying factors were estimated from multivariate regressions. HCWs showed significant increases in levels and abnormal rates of BMI, blood pressure, plasma glucose, and total cholesterol after the pandemic. Post-pandemic increases in BMI, waist circumference, and blood pressure were higher in females than in males. Workers with higher levels of education or longer job tenure had greater increases in BMI, triglyceride, and total cholesterol levels than other workers. Females had a higher incidence of abnormal BMI and hypertension than males (adjusted odds ratios [AORs] of 8.3 and 2.9, respectively). Older workers’ incidence of hypertension was higher than younger workers’ (AOR = 3.5). Preventive strategies should be implemented for HCWs susceptible to cardiovascular diseases during emerging infectious disease outbreaks.

## 1. Introduction

Since the outbreak of COVID-19, a number of studies have evaluated the workloads of healthcare workers (HCWs) during the pandemic and explored its impact on their mental health. The incidence of mental health disorders, such as depression, anxiety, and burnout, among healthcare professionals increased during the pandemic [1,2]. Work stress is a psychological syndrome of self-reported physical and mental stress in a job [3]. It can cause emotional exhaustion, depersonalization, and a decline in personal accomplishment, as it has especially during the pandemic [4,5,6]. During this time, HCWs have shown higher work stress compared to the general population [7,8]. The work stress of medical workers is a major problem in the healthcare industry, affecting not only the health of individuals, but also the quality of medical care [9]. Chronic exhaustion caused by work stress or burnout can eventually increase the risk of cardiovascular diseases (CVDs) [10]. As their total workplace health burden is related to the cardiovascular risk of HCWs, specific workplace intervention strategies are urgently needed [11]. 

Taiwan experienced its first large COVID-19 community outbreak after mid-May in 2021, later than most countries. The average daily confirmed cases in the community increased from single-digit figures to hundreds after the outbreak. When the pandemic wave began, the Central Epidemic Command Center issued several Level 3 control alerts. The measures implemented led to the development of hospital preparation and emergency response plans, which included patient volume reductions and restrictions on non-urgent examinations, surgeries, and visiting. To increase the number of hospital beds, healthcare providers were required to set up negative pressure isolation wards in hospitals and strain hospitals, expand special wards, plan for single room admissions in independent areas in hospitals with emergency responsibilities, start a responding hospital network for treatment of patients diagnosed with COVID-19, and fully empty the beds in the responding hospitals. After two months, the average daily confirmed cases declined to lower than 20. The Level 3 alert was officially in place from 19 May 2021 to 26 July 2021 [12].

It is not surprising that the COVID-19 pandemic has worsened the quality of life of HCWs by aggravating pre-existing problems. Previous studies have investigated burnout and the mental health of healthcare workers during the pandemic [5,6]. However, few studies have linked this relationship to chronic physiological illnesses such as CVDs. A multicenter longitudinal project has begun for exploring the development of psychosocial, cardiovascular, and immune markers in HCWs with different levels of COVID-19 exposure [13]. The aim of our study was to assess the possible changes in cardiovascular risk factors in HCWs and other hospital workers during the first wave of the COVID-19 pandemic in Taiwan and to identify vulnerable groups at a higher risk of increased adverse cardiovascular conditions.

## 2. Subjects and Methods

### 2.1. Study Subjects

The study hospital, which has 300 beds, is a regional teaching hospital located in New Taipei city, the biggest city in Taiwan. This study was designed as a repeated cross-sectional study, in which data from the annual employee medical check-up program were collected and examined. Excluding subjects who were pregnant (15 persons) or had missing health examination data (10 persons), we included 883 adults (approximate 97% of all employees) employed by the study hospital from 2019–2021. Of these, 514 subjects underwent physical examinations and laboratory testing once before and once after the period of the Level 3 COVID-19 alert during the study period. In addition to height and weight, we collected the workers’ biophysical and biochemical examination results, which included total cholesterol and the five indicators of metabolic syndrome (MS): waist circumference, blood pressure, fasting plasma glucose level, triglyceride level, and high-density lipoprotein cholesterol level. Using a questionnaire, we also collected data related to their sociodemographic characteristics: gender, date of birth, job tenure, education, lifestyle habits (hours of daily sleep, drinking, and smoking), and labor overwork level. Because the use of these data entailed various ethical issues, the protocol had to first receive approval from the Institutional Review Board of Fu Jen University, where the research was conducted, before the data were obtained and analyzed. The sample consisted of 60 physicians, 196 members of the nursing staff, 72 medical technicians, and 186 non-medical workers. The medical technicians group included non-physician/nurse caregivers, dietitians/nutritionists, respiratory therapists, occupational therapists, radiologic technologists, and medical technologists. The requirements of the Institutional Review Board for personal privacy prohibited the study from collecting each subject’s professional title, their job title, or the name of the department where they served.

### 2.2. Research Variables

Each study subject provided a blood sample after fasting for at least eight hours. Their blood biochemical parameters were determined after clinical tests were run in an accredited laboratory of the participating hospital. Blood biochemical cardiovascular risk factors, including serum triglyceride, plasma glucose, and cholesterol levels, were measured with a spectrophotometric autoanalyzer (Hitachi 008 Modular, Naka, Japan), and blood pressure was measured using a sphygmomanometer.

The definition of MS in this study follows the criteria of the Health Promotion Administration of the Taiwan Ministry of Health and Welfare. A person is diagnosed with MS when three or more of the five MS indicators are within abnormal ranges: waist circumference of ≥90 cm for men and ≥80 cm for women; fasting plasma glucose level of ≥100 mg/dL (5.55 mmol/L); systolic blood pressure of ≥130 mmHg and diastolic blood pressure of ≥ 85 mmHg; triglyceride level of ≥150 mg/dL; and HDL cholesterol level of ≤40 mg/dL in men or ≤50 mg/dL in women. A person with a blood pressure reading of ≥140/90 mmHg was classified as hypertensive [14]. According to the criteria of the Department of Health in Taiwan, the subjects were divided into three subgroups based on their body mass index (BMI; kg/m^2^): obese (BMI ≥ 27), overweight (24 ≤ BMI < 27), and normal (BMI < 24) [15]. We compared the group of subjects with a BMI of ≥24 with the normal group out of consideration for the sample size. Then, whether the values were within an abnormal range was determined, followed by an analysis of the distribution of the demographic characteristics.

The sociodemographic characteristics analyzed in this study were the subjects’ demographic information and lifestyle habits, including drinking, smoking, and average hours of sleep during the previous one month. The classified variables were then redefined according to the subject’s responses, such as smoking (yes/no), drinking (yes/no), hours of sleep (>7 h/≤7 h), and job tenure (<10 years/≥10 years). Subjects were asked about work stress only during the post-pandemic examination to show the level of overwork during the pandemic. This study used a scale that measures overwork in relation to the individual and the job, developed by the Taiwan Institute of Labor and Occupational Safety and Health, for evaluating each subject’s work stress [16]. According to the stratification of this scale, individual-related overwork levels were categorized into three subgroups: mild (<50), medium (50–70), and severe (>70). Job-related overwork levels were also categorized into three subgroups: mild (<45), medium (45–60), and severe (>60). As only 20 workers reported severe overwork, the overwork levels of the study subjects were categorized into two subgroups: mild, and medium to severe.

### 2.3. Statistical Analysis

The descriptive statistics of each biophysical and biochemical parameter were calculated based on the subjects’ two health examinations (i.e., the pre-pandemic examination and post-pandemic examination). In this study, the biophysical and biochemical parameters were categorized as binary or continuous variables. The Shapiro–Wilk Test was used to test the normal distribution. As their distributions were not skewed, the differences in individual biophysical and blood biochemical parameters before and after the pandemic were tested using paired *t*-tests. McNemar’s test was used to assess the differences in the frequency distributions of individual MS factors, abnormal BMI, and biophysical and blood biochemical parameters between the two examinations. The study defined an incidence of adverse outcome as an abnormal level of a parameter measured in the post-pandemic examination that had been normal in the pre-pandemic examination. Using workers with no cardiovascular incidents as a reference, the relative risk of adverse cardiovascular incidences was estimated with the adjusted odds ratio (AOR), which was calculated from the regression coefficients of multivariate logistic regression models after adjusting for covariates. With the post-pandemic examination–pre-pandemic examination differences as dependent variables, a linear regression model and dummy variable approach were used to deal with continuous variables. The variance inflation factor was also calculated for each model to prevent unreliable estimates of coefficients with possible high correlations between predictors. SPSS version 22.0 (IBM Corp., Armonk, NY, USA) was used for the statistical analyses. Statistical significance was set at *p* ≤ 0.05.

## 3. Results

Table 1 shows the frequency distributions of sociodemographic characteristics. A higher proportion of subjects were female (83%), educated at college level or above (82%), non-smokers (96%), and non-drinkers (65%), and obtained less than seven hours of sleep per night (76%). Twenty-six percent of the subjects reported medium to severe overwork related to their job during the pandemic. After excluding age and job tenure, no significant differences between the two examinations were found in the other sociodemographic characteristics (data not shown). Therefore, we used the characteristics filled out in the post-pandemic examination questionnaire as the underlying risk factors in further analyses.

Table 2 shows the distribution of biophysical and blood biochemical parameters from each examination. Significant increases in the levels of BMI, blood pressure, plasma glucose, and total cholesterol were observed between examinations. Table 3 shows significant increases in the abnormal rates of blood pressure, plasma glucose, and total cholesterol after the pandemic. The differences in the changes in biophysical and biochemical parameters between the two examinations by sociodemographic characteristics are shown in Table 4. Female workers had greater increases by 0.55 kg/m^2^, 1.50 cm, and 6.63/5.31 mmHg (*p* < 0.001) in BMI, waist circumference, and systolic/diastolic blood pressure, respectively, after the pandemic compared to males. Workers with a higher level of education (equal to or above the college level) showed greater increases by 0.40 kg/m^2^ (*p* = 0.024), 0.91 cm (*p* = 0.046), 17.35 mg/dl (*p* = 0.020), and 13.20 mg/dl (*p* < 0.001) in BMI, waist circumference, triglyceride level, and total cholesterol level, respectively, compared to their counterparts with a lower level of education. Workers with a longer job tenure (≥ 10 years) showed a greater increase, by 9.34 mg/dl (*p* < 0.001), in their total cholesterol level than those with a shorter tenure. Smokers showed an increase greater by 1.04 and 2.49 cm in BMI and waist circumference, respectively, than that of non-smokers.

Table 5 shows the AORs for the likelihood of increases in MS, abnormal MS factors, and total cholesterol after the pandemic. The female workers’ AOR was 8.3 (*p* = 0.043) and 2.9 (*p* = 0.036) for incidence of abnormal BMI and hypertension, respectively, in the post-pandemic examination. Compared to the younger age groups (<40 years), the older age groups (> 50 years) had an AOR of 3.6 (*p* = 0.003) for incidence of hypertension. The incidences of MS after the pandemic were not associated with the underlying risk factors. Greater individual-related overwork negatively affected (AOR = 0.3) the risk of incidence of hypertension at the post-pandemic examination (*p* = 0.003). Greater job-related overwork positively affected (AOR = 2.3) the risk of incidence of hypertension, but the association was not significant (*p* = 0.104). Smoking, drinking, and sleep duration were also not related to the increased risk of hypertension. The factors in the multivariate regression models were not collinear.

## 4. Discussion

The present study found that HCWs had an increased risk of developing CVDs, especially hypertension, as well as an increased risk of developing diabetes and dyslipidemia in the period following the COVID-19 pandemic. Our findings reveal that HCWs may not only suffer from stress but also be at increased cardiovascular risk during the pandemic. Since HCWs are crucial to the functioning of the healthcare system, their health is of the highest concern and should not be overlooked. Medical workers may be a population that is particularly vulnerable to mental illness due to long working hours, the risk of infection, the lack of personal protective equipment, physical fatigue, and separation from their family members during critical times [8]. Job stress has been recognized as a risk factor in several adverse health outcomes, mainly CVDs [17]. In addition, several studies have suggested that overwork is a risk factor in acute myocardial infarction [18] and hypertension [19]. Belkic et al. showed that work stress can interfere with the neuroendocrine system and result in sympathetic nervous system hyperreactivity [17]. Sawai et al. [20] found that mental stress has an influence on the plasma homocysteine level and blood pressure variation. A recent hospital-based study from India reported that work stress was strongly correlated with the blood lipid profile and blood pressure of 40 HCWs aged 25–40 years who had direct contact with COVID-19 patients [21]; the authors considered physical and psychological stress caused by the COVID-19 assignment as a risk factor in increased triglyceride and low-density lipoprotein levels and decreased high-density lipoprotein levels. Alameri et al. used electronic surveys in hospitals and healthcare institutions in Abu Dhabi for a cross-sectional investigation and reported correlations between burnout severity and cardiovascular risk in healthcare professionals during the COVID-19 pandemic [22]. Although the causal effects were weak due to the use of cross-sectional design, the authors concluded that healthcare practitioners with burnout and emotional exhaustion have an elevated cardiovascular risk, and they implied that the increased cardiovascular risk of HCWs may be related to the high psychological and physical stress caused by the extra burden of their pandemic duties or by the pandemic environment. A review study suggested several measures, including physical activity, a balanced diet, good sleep hygiene, family support, meaningful relationships, reflective practices, and small-group discussions, to prevent or reduce burnout among HCWs in stressful situations [23]. More evidence of the association between COVID-19-related burnout and cardiovascular risk is necessary in order to develop preventive measures to further reduce HCWs’ risks of CVDs.

Our analysis was based on repeated health check-up data, which can be very useful in assessing cardiovascular risk and its factors among hospital workers, since they are obtained from standardized laboratory tests. However, overwork related to the effort to combat COVID-19 cannot be linked to the incidence of CVDs in our study. The protective effects of individual-related overwork on hypertension should be further investigated, also. The majority of participants in the study were not overworked. Workers may experience disparities in workloads even during busy periods. In order to meet the demand for health care services, some non-frontline employees were required to work remotely, which may have changed their daily routines. This may have resulted in confusion among participants regarding their acknowledgement of work stress related to individuals or work. Moreover, because we asked subjects about their personal overwork level only in the post-pandemic examination, we could not investigate whether each HCW’s work burden or stress level had been altered during the period of study. Therefore, whether the increased cardiovascular risk was caused by burnout due to the pandemic remains unclear in the present study. Ensuring that interventions are theoretically designed to address the occupational determinants of stress and that workers are involved in change processes should increase the likelihood of better health outcomes for HCWs during emergencies [24]. A follow-up and moderated approach are needed to establish the cause–effect relationships between the event, behavior, and health outcomes.

It is helpful to identify susceptible groups in order to develop specific strategies to prevent the risk of CVDs and stress among health professionals. The present study shows that the relationship between the pandemic and increased blood pressure and BMI is more apparent in female HCWs than in male workers. Previous studies have shown that the effect modification by age of the relationship between overweight and cardiovascular risk is less apparent in female HCWs than in male workers, implying that the control of body weight is even more important for female HCWs [25]. A longitudinal study reported that middle-aged women may gradually lose some of the protective effects of estrogen as they proceed through menopause, thus making them more susceptible than men to the effects of exposure to CVD risk factors [26]. Taylor et al. suggested that women have a heightened biological sensitivity to socioeconomic status [27]. Being a woman was also associated with higher levels of compassion fatigue, emotional exhaustion, and depersonalization [28,29]. During the pandemic, working women were burdened with more household duties and caring for children. This could have adverse effects on their health as well [30,31]. Therefore, weight loss programs to lower hypertension seem to be crucial for female workers. On the other hand, professional specialism has been found to be related to emotional exhaustion and depersonalization among HCWs during the COVID-19 pandemic [32,33]. Female workers were in the majority in the current study sample, which raises the question of what their professions were in the workplace. In our study, the proportion of the physician, nurse, technician, and non-medical worker groups in female workers were 4.4%, 44.7%, 12.7%, and 38.2%, respectively. Few female physicians were observed. Lluch et al. reviewed 75 studies and found that female gender, nursing, and the workplace attending to COVID-19 patients were the critical factors that influenced personal quality of life during the pandemic [34]. Burnout was found to be worse in nurses than in doctors and other health workers [35].However, another study suggested that nurses and therapists are less likely to show compassion fatigue and burnout compared to physicians and psychologists [36]. Self-care, organizational justice, and implementation of individual and organizational preventive strategies during emerging infectious disease outbreaks were successful in protecting HCWs from developing emotional exhaustion [37]. Therefore, a program of effective psychological and coping strategies is likely to provide substantial benefits in preventing underlying diseases in susceptible groups.

Age, not surprisingly, plays a major role in increasing the odds of developing CVDs. A study from Taiwan reported that older age was related to worse self-rated health, and age showed a reverse-U-shaped relationship with psychological health [38]. A review study showed that older workers in physically demanding jobs experience greater irritability, and that the effects could be moderated by type of occupation and gender. The authors also suggested that organizations should establish measures to compensate for age-related losses in physical capacity [39]. In addition to higher levels of education, the results of the present study indicated that longer job tenure, independent of age, could increase triglyceride and total cholesterol levels during the pandemic. This may be because senior workers or those with a higher level of education may be more likely to be assigned to managerial roles with greater responsibilities, especially during emergencies. A Taiwanese study found that managers have higher risks of MS and CVDs [40]. The authors also suggested that these managers might have been promoted due to their own professional expertise, so might endure more work-related stress and health risks. If so, it is worth examining whether certain factors, other than socioeconomic status, create these theoretical differences. An African study found that, despite higher levels of education and physical activity, MS was more prevalent among technicians who had a specific expertise than schoolteachers [41]. Hospital managers should pay greater attention to employees with particular skills even if their work responsibilities are taken for granted during critical periods.

The effects of smoking on mental health and CVDs are well known. In addition, the relationship between emotional exhaustion and sleep quality in healthcare professionals during the COVID-19 pandemic has already been studied [22,42,43,44]. Sleep quality can be influenced by high psychological distress, high emotional exhaustion, depersonalization, and low personal accomplishment [45]. Short sleep durations and poor sleep are increasingly being linked to the development of CVDs [46,47]. However, the present study failed to determine the relationship between increased cardiovascular risk and adverse habits or short sleep duration. In addition to the variation in definitions between studies, our findings of lack of differences in individual behaviors before and after the pandemic may imply that there were few changes in daily life due to the pandemic, which thus resulted in a different conclusion from those of other studies. However, we did not consider other health behaviors, including exercise and diet, which may also bias our findings. A French study claims that it was difficult to promote healthy behaviors like physical activity and healthy diets during the pandemic [48]. Future investigations should include observations on individual lifestyles over time to explain longitudinal effects.

The use of pre-/post-test difference comparisons with a dependent sample in this study is likely to reduce confounding effects caused by individual sociodemographic characteristics. Nonetheless, our analysis also had several limitations that should be considered. First, because of the failure to collect the professional and job titles of subjects due to the requirements of the IRB, we could not further explore the impacts of occupational factors on the risks of MS and CVDs during the pandemic. Moreover, because we did not ask the subjects about their personal job assignments, e.g., whether they were involved in the front line in caring for patients with COVID-19, the association between work stress caused by pandemic-related changes in working conditions and physiological health could not be evaluated. In addition, the variability among institutions should be noted. Even among hospitals with COVID-19 responsibilities, the burdens of responding to the pandemic are not the same. Our findings from one such hospital cannot be extrapolated to other medical settings. Finally, although medication use was initially considered, we did not include this factor as we did not have sufficient information about the subjects’ prior histories of hypertension, diabetes, or hyperlipidemia. Consequently, we could not interpret the study results in specific ways because the subjects were likely to use certain medications for comorbidities, which could have biased the results. Additional studies using a multicenter approach with more variables may be able to reflect in more detail on independent relationships.

## 5. Conclusions

This study found that hospital workers’ risks of hypertension, diabetes, and dyslipidemia increased after the COVID-19 pandemic. Such an increase might be further magnified by specific sociodemographic factors, especially female gender, workplace seniority, and higher levels of education. Preventive strategies for CVDs should be specifically designed as interventions for susceptible groups during emerging infectious disease outbreaks. Longitudinal check-up data should be collected and analyzed regularly to enhance the utility of check-up examinations for detecting high-risk groups of HCWs. Future investigations should explore the relationships between job assignment, burnout, and cardiovascular risk, which may suggest effective work shifts and health promotion programs to reduce the cardiovascular risk during the pandemic.

## Figures and Tables

**Table 1 ijerph-19-16114-t001:** Sociodemographic characteristics of study subjects during the period before the first pandemic wave.

Characteristic	N	%
Sex		
Male	86	16.73
Female	428	83.27
Age		
<40 years	200	38.91
40–50 years	180	35.02
>50 years	134	26.07
Education *		
High school	77	14.98
College or above	423	82.3
Job tenure		
<10 years	263	51.17
≥10 years	251	48.83
Smoking		
No	493	95.91
Yes	21	4.09
Drinking		
No	334	64.98
Yes	180	35.02
Sleep duration		
>7 h	125	24.32
≤7 h	389	75.68
Job overwork		
Mild (<45)	379	73.74
Medium to severe	135	26.26
Individual overwork		
Mild (<50)	378	73.54
Medium to severe	136	26.46

Note: * 14 observations were missing. Before and after the pandemic, smoking, drinking, and sleep duration did not differ significantly.

**Table 2 ijerph-19-16114-t002:** Differences in the levels of biophysical and biochemical parameters between pre-pandemic examination and post-pandemic examination.

Biophysical or Biochemical Parameter	Pre-Pandemic Examination			Post-Pandemic Examination			Paired *t*-Test	
	Median	Mean	SD	Median	Mean	SD	*t*	*p*-Value
Body mass index (kg/m^2^)	22.89	23.96	4.651	23.08	24.19	4.691	3.95	<0.001
Waist circumference (cm)	78	79.27	11.425	77.25	79.02	11.544	−0.94	0.348
Systolic blood pressure (mmHg)	122	123.52	15.851	123	124.81	15.999	2.08	0.038
Diastolic blood pressure (mmHg)	73	73.17	11.302	75	75.27	11.03	4.42	<0.001
Plasma glucose (mg/dL)	92	95.47	17.5	93	97.49	23.639	2.69	0.007
Triglycerides (mg/dL)	75	94.13	62.717	81	97.17	72.765	1.222	0.222
High-density lipoproteins (mg/dL)	62	61.8	14.682	61	62.14	15.188	0.908	0.364
Total cholesterol (mg/dL)	184	185.83	31.795	189	192.6	33.715	6.121	<0.001

**Table 3 ijerph-19-16114-t003:** Pairwise difference in abnormal biophysical and biochemical parameters between pre-pandemic examination and post-pandemic examination.

Biophysical or Biochemical Parameter	Pre-Pandemic Examination		Post-Pandemic Examination		McNemar Test	
	Abnormal (n)	Rate (%)	Abnormal (n)	Rate (%)	X^2^	*p*-Value
Body mass index	206	40.08	212	41.25	0.521	0.47
Metabolic syndrome	88	17.12	99	19.26	1.695	0.913
Waist circumference	185	35.99	177	34.44	0.598	0.44
Systolic blood pressure	55	10.7	73	14.2	3.083	0.051
Diastolic blood pressure	26	5.06	43	8.37	5.02	0.025
Plasma glucose	119	23.15	138	26.85	4.32	0.038
Triglycerides	74	14.4	60	11.67	2.561	0.11
High-density lipoproteins	90	17.51	93	18.09	0.066	0.798
Total cholesterol	150	29.18	206	40.08	26.078	<0.001

**Table 4 ijerph-19-16114-t004:** Difference in changes in levels of biophysical and biochemical parameters between pre-pandemic examination and post-pandemic examination by sociodemographic characteristics.

Characteristic	Body Mass Index		Waist Circumference		Systolic Blood Pressure		Diastolic Blood Pressure		Plasma Glucose		Triglycerides		High–Density Lipoproteins		Total Cholesterol	
	b	*p*-Value	b	*p*-Value	b	*p*-Value	b	*p*-Value	b	*p*-Value	b	*p*-Value	b	*p*-Value	b	*p*-Value
Sex (vs. Male)																
Female	0.553	0.001	1.496	<0.001	6.627	<0.001	5.308	<0.001	0.349	0.872	4.782	0.49	–0.333	0.755	4.628	0.14
Age (vs. < 40 years)																
40–50 years	–0.195	0.203	–0.554	0.161	–1.514	0.354	–0.553	0.662	–3.459	0.089	–0.023	0.997	0.248	0.804	–2.421	0.409
> 50 years	–0.107	0.553	–0.293	0.532	2.254	0.245	–0.157	0.916	–1.431	0.553	5.788	0.451	0.104	0.93	–2.025	0.56
Education (vs. High school)																
College or above	0.399	0.024	0.908	0.046	2.399	0.203	0.648	0.657	0.38	0.871	17.353	0.02	–0.693	0.546	13.201	<0.001
Job tenure (vs. < 10 years)																
≥10 years	0.261	0.056	0.607	0.085	1.974	0.177	0.818	0.469	1.644	0.365	–5.588	0.334	1.575	0.078	9.338	<0.001
Smoking (vs. No)																
Yes	1.036	0.001	2.494	0.001	0.562	0.859	2.984	0.225	–0.182	0.963	0.695	0.956	2.288	0.238	7.991	0.161
Drinking (vs. No)																
Yes	–0.243	0.054	–0.580	0.075	1.23	0.361	1.515	0.146	1.429	0.393	2.731	0.608	–1.211	0.141	–2.109	0.382
Sleep duration (vs. > 7 h)																
≤7 h	–0.202	0.135	–0.518	0.138	–1.096	0.448	–0.005	0.996	2.363	0.189	–1.514	0.791	–1.041	0.238	1.51	0.56
Individual overwork (vs. Mild)																
Medium or severe	0.075	0.7	0.148	0.767	0.885	0.67	–1.285	0.424	–1.621	0.53	–15.416	0.061	–1.131	0.373	–3.814	0.306
Job overwork (vs. Mild)																
Medium or severe	–0.125	0.521	–0.106	0.834	–2.069	0.322	0.61	0.706	–0.973	0.708	7.366	0.373	2.161	0.09	4.343	0.246

**Table 5 ijerph-19-16114-t005:** Risk of incidence of abnormal biophysical and biochemical parameters after the COVID-19 pandemic by sociodemographic characteristics.

Characteristic	Body Mass Index		Metabolic Syndrome		Waist Circumference		Hypertension		Plasma Glucose		Triglycerides		High-Density Lipoproteins		Total Cholesterol	
	AOR	*p*-Value	AOR	*p*-Value	AOR	*p*-Value	AOR	*p*-Value	AOR	*p*-Value	AOR	*p*-Value	AOR	*p*-Value	AOR	*p*-Value
Sex (vs. Male)																
Female	8.296	0.048	0.528	0.151	0.645	0.091	2.906	0.036	0.783	0.558	0.458	0.101	2.707	0.193	2.073	0.067
Age (vs. < 40 years)																
40–50 years	2.253	0.131	1.063	0.9	1.583	0.066	1.35	0.45	1.625	0.25	1.193	0.745	1.245	0.66	1.193	0.597
>50 years	2.788	0.108	1.152	0.795	1.449	0.21	3.57	0.003	1.646	0.304	1.411	0.571	1.109	0.867	1.694	0.162
Education (vs. High school)																
College or above	2.036	0.301	0.747	0.579	0.868	0.62	1.034	0.932	1.427	0.475	4.371	0.164	0.908	0.866	1.738	0.145
Job tenure (vs. <10 years)																
≥10 years	0.496	0.135	0.668	0.34	0.739	0.173	0.628	0.153	0.993	0.985	0.788	0.618	0.726	0.477	1.586	0.113
Smoking (vs. No)																
Yes	3.288	0.162	2.223	0.249	0.744	0.55	0	0.998	0.542	0.564	0.84	0.872	0.849	0.881	1.368	0.639
Drinking (vs. No)																
Yes	1.196	0.682	0.736	0.447	1.009	0.965	1.289	0.405	0.713	0.342	0.763	0.556	1.035	0.934	0.728	0.253
Sleep duration (vs. >7 h)																
≤7 h	2.729	0.111	1.5	0.386	1.295	0.253	1.221	0.559	0.704	0.304	1.657	0.366	1.129	0.789	1.149	0.632
Individual overwork (vs. Mild)																
Medium or severe	0.322	0.122	0.631	0.46	0.961	0.9	0.308	0.03	0.59	0.332	0.53	0.342	1.562	0.473	0.734	0.471
Job overwork (vs. Mild)																
Medium or severe	1.749	0.391	1.352	0.618	1.436	0.256	2.257	0.104	1.652	0.337	2.252	0.199	0.396	0.179	0.989	0.98

## Data Availability

The data sets generated and analyzed during the current study are not publicly available due to privacy/ethical restrictions.

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
