# Peer review of "Cardiovascular Risk Factors in Hospital Workers during the COVID-19 Pandemic: A Hospital-Based Repeated Measures Study"

_ijerph, 2022, doi:10.3390/ijerph192316114_

Round 1

Reviewer 1 Report

Subjects and Methods Section:

1.       Please clarify the method of estimation the sample size or power test analysis.

2.       The reference of definition of MS and BMI norms are missing. It should be added.

Results Section:

3.       In the table 2 the BMI unit is missing.

Discussion section:

4.       Your study revealed, that neither individual overwork (except negative affection on hypertension, which has not been clarified) nor job overwork has no impact on examined CVD risk factors. Most of the individuals included to the study were not overworked (73%). It should be emphasized and expanded in the Discussion Section.

5.       Was there any clinical significance of your results? Please describe the clinical implication of your study.

6.       The limitation of the study were accurately formulated ( lack of professional titles, lack of data of workload or stress level before pandemic wave, lack of lifestyle data such as smoking, drinking, sleep duration before pandemic wave, and lack of data about diet or medication use) and their multitude has a negative impact on the quality of the study.

Reviewer 2 Report

Dear Editor, dear authors,

I enjoyed reading this article. The article is well-written but has some limitations.

A summary

The paper is focused on the examination of CVD risk factors before and after the first sizeable COVID pandemic wave in a Taiwan hospital. The paper's main contribution is that it is a rare study examining CVD factors during the pandemic.

General concept comments

The title of the article could be more precise. It leads to two significant comments:

The study population included 36% of non-HCW. Therefore, the HCW should not be used and replaced appropriately in the title and manuscript.

The authors did not examine cardiovascular risks, which was surprising as they have all data to calculate it (Framingham, for example), but cardiovascular risk factors. Therefore, the title and the study aim should be changed accordingly.

Introduction

The sentence "The measures implemented led to the development of hospital preparation and emergency response plans, which included outpatient volume load reduction; restrictions on patients, accompanying (visiting) relatives, and friends and non-urgent examinations; and suspension of surgical procedures" is too long and hard to read. Please, rewrite this sentence.  

Subjects and methods

The sentence "As a hospital with COVID-19 responsibilities the study hospital enforced emergency response plans and treated 24,634 individuals screened with the polymerase chain reaction test for COVID-19 and 105 hospitalizations for confirmed severe disease during the period of the Level 3 alert" is not necessary as it not referred to your study population.

The study sample included 186 (36%) out of 514 non-HCW – therefore, the paper could be referred to only HCW as well as the general conclusion of the paper.

The information about the study site needs to be included. In which hospital/s data were collected?

Statistical analysis

The sentence" As their distributions were not skewed… using paired t-tests" – did you mean that data followed the normal distribution? If so, you did not mention that you use the Kolmogorov-Smirnov Test, Shapiro-Wilk Test, or Anderson-Darling test to test the normal distribution of data.

Results

I suppose that Table 1 refers to the period before the first pandemic wave – you should add this detail.

It would be interesting to see in Table 1, or under the table as a text, whether there was a change in variables smoking, drinking, sleep duration, job overwork, or individual overwork at the second point of measurement.

In sentence line 166, the measure of BMI is missing (0.55 kg/m2), and in line 169, (0.4 kg/m2).

Tables 2, 3, and 4 go one after another. Please add some text between them. It is hard to follow the results in this way.

The title of the Table 3 could be more precise as it depicts differences in frequencies using the McNemar. Please, add a more specific title.

Discussion

The first sentence is not necessary. It is optional, as you explain the same in the introduction section but referenced with different references (1 and 2 in the introduction section and 5 and 6 in the discussion section). Please, check the references.

Lines 255 and further – please check and include other references. Women had to care for households and children more during the pandemic. This fact could also be a reason for their worsening health condition.

Conclusion
The conclusion is decent. The conclusion needs to be adjusted when the manuscript is changed according to the comments.

Round 2

Reviewer 2 Report

The authors did all necessary changes in the manuscript.